# Sports instructors' job insecurity and turnover intention in South Korea: A moderated mediation model of abusive supervision, work engagement, and perceived organizational support

Kwon-Hyuk Jeong[ID][1⊙], Heesu Mun[ID][2⊙*], Geon-Ha Jeong[1*⊙]

**1** Department of Taekwondo, College of Physical Education, Kyung Hee University, Yongin-si, Gyeonggi-do, Republic of Korea, **2** Department of Physical Education, Graduate School of Education, Kyung Hee University, Yongin-si, Gyeonggi-do, Republic of Korea

⊙ These authors contributed equally to this work.
* heesumun@khu.ac.kr (HM); jgh1033@khu.ac.kr (GHJ)

## Abstract

This study proposes a moderated mediation model examining the impact of job insecurity on turnover intention among non-regular sports instructors in South Korea, through the mediating effect of abusive supervision and the moderated mediation effects of work engagement and perceived organizational support. Analysis of survey data from 267 participants using SPSS and AMOS revealed that job insecurity significantly and positively influenced turnover intention and abusive supervision, with abusive supervision demonstrating a partial mediating effect on the relationship. Furthermore, both work engagement and perceived organizational support exhibited significant moderated mediation effects, such that the mediated pathway strengthened when these factors were low. These findings underscore the necessity of leadership training, psychological support enhancement, and stable employment policies to mitigate employment instability in the sports industry, contributing to human resource management theory in sports management based on JD-R theory and conservation of resources theory.

## 1. Introduction

The sports industry has expanded rapidly, generating both employment opportunities and economic value across many countries [1]. As the sector has grown, demand for sports instructors has also increased in both public and private settings, including community sports facilities, school sport programs, and local government initiatives [2–4]. Sports instructors play an important role in promoting physical activity and in improving participants' physical health, quality of life, and mental well-being [5]. Despite these contributions, many sports instructors and coaches work under poorly

**Data availability statement:** The datasets generated and analyzed during the current study are publicly available via the Open Science Framework (OSF) repository (https://osf.io/92wsf/).

**Funding:** The author(s) received no specific funding for this work.

**Competing interests:** The authors have declared that no competing interests exist.

defined employment conditions characterized by short-term contracts, performance-based evaluation, and excessive job demands [6–7].

Non-regular sports instructors, in particular, are likely to experience elevated job stress and stronger turnover intention than regular employees because they often face unstable employment, limited promotion opportunities, and insufficient welfare benefits [8–9]. Prior research has shown that perceived job insecurity is associated with poorer mental health, lower organizational commitment, and stronger intentions to leave one's job [10–11]. In the sports sector, these concerns are especially important because employment instability may undermine not only employee well-being but also service continuity and program quality.

Among the organizational factors that may shape this process, abusive supervision is particularly relevant. Supervisors play a central role in structuring communication, trust, and work relationships in sport organizations [12]. Abusive supervision refers to sustained hostile verbal and nonverbal supervisory behavior, excluding physical contact [13]. In insecure work environments, employees may be more sensitive to supervisory mistreatment, and strained supervisor-subordinate relationships may be more likely to develop [14]. Accordingly, abusive supervision may help explain how job insecurity is linked to turnover intention among non-regular sports instructors.

This study also considers two important resources: work engagement and perceived organizational support. Work engagement, typically characterized by vigor, dedication, and absorption, reflects a positive and fulfilling work-related psychological state [15]. Perceived organizational support refers to employees' beliefs that the organization values their contributions and cares about their well-being [16]. Both constructs may weaken the negative association between job insecurity and turnover intention by buffering the adverse effects of insecure and stressful work conditions.

Despite a substantial body of research on job insecurity across a wide range of occupations, relatively limited attention has been paid to how job insecurity operates in the sport management context [6]. In particular, prior studies have more often examined direct associations or simple mediating pathways than moderated mediation processes among sports instructors [17–19]. This omission is important because sports instructors occupy a distinctive organizational position characterized by close supervisory dependence, emotional labor, and unstable employment arrangements. The present study addresses this gap by examining whether abusive supervision mediates the association between job insecurity and turnover intention, and whether this indirect association varies according to levels of work engagement and perceived organizational support.

This context is especially meaningful in South Korea, where non-regular employment remains widespread and is often accompanied by disparities in job security, welfare benefits, and organizational protection [20–23]. For non-regular sports instructors, such structural insecurity may intensify the psychological consequences of precarious work and increase the importance of supervisory and organizational resources. In this sense, the present study extends prior job insecurity research to an underexplored sport management setting and refines the application of the Job Demands-Resources

(JD-R) model and Conservation of Resources (COR) theory by examining how personal and organizational resources buffer the indirect association between job insecurity and turnover intention in a precarious employment context.

Accordingly, this study examines the association between job insecurity and turnover intention among non-regular sports instructors in South Korea, focusing on the mediating role of abusive supervision and the moderating roles of work engagement and perceived organizational support. By doing so, the study aims to contribute to sport management scholarship and to provide practical implications for improving employment sustainability and retention in sport organizations.

## 2. Theoretical background

### Employment structure and institutional context of sports instructors in South Korea

The employment structure of sports instructors in South Korea is shaped by policy priorities, budget allocations, and institutional arrangements [20]. Sports instructors work across diverse settings, including community programs, schools, youth sport initiatives, and publicly supported sport promotion organizations [24]. Many of these positions are contract-based and renewed annually, with renewal often depending on ambiguous performance criteria and shifting policy directions [25]. As a result, job insecurity may become a persistent feature of the employment environment rather than a temporary concern [26].

In addition, non-regular sports instructors often face employment conditions that differ substantially from those of regular employees. These disparities may include limited eligibility for social insurance, insufficient recognition of prior work experience, unclear performance evaluation systems, and weak union representation [27]. Such conditions can undermine employment stability and may heighten perceptions of vulnerability within the workplace [28].

Since 2020, the South Korean government has introduced several measures intended to improve the treatment of sports instructors [29]. However, sports instructors have not been fully incorporated into broader initiatives aimed at regularizing non-regular workers, and the extent of policy protection remains unclear [30]. In addition, the decentralization of community sports policy and the time-limited nature of some budget allocations may further weaken employment stability at the local level [31]. These institutional features make South Korean non-regular sports instructors a particularly relevant population for examining how employment precarity is associated with supervisory strain and turnover intention.

Accordingly, job insecurity among non-regular sports instructors should be understood not only as an individual concern but also as an institutional and organizational issue. It may affect service quality, organizational stability, and the broader sustainability of public and community sport systems.

### Job insecurity theory in the sports industry

Job insecurity refers to employees' perceived threat to the continuity and stability of their employment [22,32]. It reflects a psychological stress response rooted in perceived uncertainty rather than in objective employment status alone [33]. Prior scholarship has distinguished between objective job insecurity, which is shaped by external employment conditions, and subjective job insecurity, which is shaped by personal perceptions, emotions, and expectations [34].

In the sports industry, job insecurity may be especially salient because many instructors work under short-term or contract-based arrangements [7,35]. These conditions may restrict future planning, weaken organizational attachment, and reduce work motivation [20,36]. Given the importance of instructor continuity and psychological well-being in sport service delivery, job insecurity may be associated not only with employee outcomes but also with broader organizational effectiveness and service quality [37–38].

### Theory of abusive supervision and its implications in sports organizations

Tepper defined abusive supervision as subordinates' perceptions of sustained hostile verbal and nonverbal supervisory behaviors, excluding physical contact [13]. This form of leadership has been linked to lower job satisfaction, weaker organizational commitment, and stronger turnover intention [39].

In sport organizations, abusive supervision may be particularly consequential because hierarchical authority structures, short-term contracts, and high emotional labor can intensify employees' dependence on supervisors [40]. Prior research suggests that when power is concentrated and employment is unstable, employees may be more vulnerable to supervisory mistreatment and its psychological consequences [41]. Recent studies have also indicated that abusive supervision may strengthen the association between job insecurity and turnover-related outcomes through emotional exhaustion and reduced organizational trust [42]. Taken together, these findings suggest that abusive supervision is an important relational mechanism through which employment insecurity may become linked to negative workplace attitudes in sport organizations.

## 3. Research hypothesis

### Relationship among job insecurity, abusive supervision, and turnover intention

Job insecurity refers to workers' perceptions of threats to employment continuity, which are associated with psychological strain and weaker organizational attachment. According to the JD-R model, high job demands may contribute to strain and withdrawal-related outcomes, including turnover intention [43]. Similarly, COR theory suggests that perceived employment instability may threaten valued resources, leading employees to consider leaving as a protective response [44]. Meta-analytic evidence has consistently shown that job insecurity is positively associated with turnover-related outcomes [45,46]. Based on this literature, the following hypothesis is proposed.

H1. Job insecurity is positively associated with turnover intention among non-regular sports instructors.

In organizational contexts characterized by high job insecurity, supervisor-subordinate relationships may deteriorate [47]. When supervisors disregard employees' psychological well-being and treat them instrumentally, such behavior may be perceived as abusive supervision. Social exchange theory suggests that limited support from the organization or supervisor may erode relational quality and contribute to negative interpersonal dynamics [48]. Prior research has further suggested that unstable employment environments may heighten perceptions of abusive supervision [49]. Thus, the following hypothesis is proposed.

H2. Job insecurity is positively associated with abusive supervision among non-regular sports instructors.

Abusive supervision reflects low-quality supervisory relationships marked by hostility, alienation, and lack of consideration. Emotional events theory suggests that negative workplace interactions may elicit adverse emotional responses and contribute to unfavorable attitudinal outcomes, including turnover intention [50]. Empirical studies have also shown that abusive supervision is positively associated with turnover intention [51]. Therefore, the following hypothesis is proposed.

H3. Abusive supervision is positively associated with turnover intention among non-regular sports instructors.

### Mediating effect of abusive supervision

Job insecurity has been associated with reduced organizational trust, lower commitment, and stronger turnover intention [32,52]. In contract-based employment contexts, such insecurity may also shape employees' perceptions of supervisory relationships. Abusive supervision may represent an interpersonal mechanism through which employment-related stress becomes linked to turnover intention. In sport organizations, where supervisors often play a central role in evaluation, coordination, and work climate, such relational processes may be especially salient [53,54]. On this basis, the following hypothesis is proposed.

H4. Abusive supervision mediates the association between job insecurity and turnover intention.

### Moderated mediation effect of work engagement

Job insecurity may be associated with stronger turnover intention through abusive supervision, but this indirect association may vary depending on employees' level of work engagement. Within the JD-R framework, work engagement can be

understood as a positive work-related resource characterized by vigor, dedication, and absorption [15,43]. Employees with higher work engagement may cope more effectively with stressful work conditions, thereby weakening the association between job insecurity and abusive supervision perceptions [55]. Accordingly, work engagement may buffer the indirect association between job insecurity and turnover intention via abusive supervision.

H5. Work engagement moderates the indirect association between job insecurity and turnover intention via abusive supervision, such that the indirect association is weaker at higher levels of work engagement.

## Moderated mediation effect of perceived organizational support

Perceived organizational support (POS) refers to employees' beliefs that the organization values their contributions and cares about their well-being [16]. From a COR perspective, POS may function as an organizational resource that helps offset the adverse effects of employment insecurity. When employees perceive stronger organizational support, the negative association between job insecurity and turnover intention through abusive supervision may be attenuated [56,57]. Therefore, POS is expected to weaken the indirect association between job insecurity and turnover intention via abusive supervision.

H6. Perceived organizational support moderates the indirect association between job insecurity and turnover intention via abusive supervision, such that the indirect association is weaker at higher levels of perceived organizational support.

Accordingly, the conceptual research model proposed in this study is presented in Fig 1.

## 4. Materials and methods

### Participants and procedure

The target population for this study was defined as Korean nationals aged 19 or older who had worked as non-regular sports instructors within the past year. Participants were recruited using convenience sampling, a non-probability sampling method, over a two-month period from April 7 to June 10, 2025, following research approval. Participants were recruited by posting posters explaining the study purpose and procedures at 11 gyms and 3 sports centers. Participants who consented to participate were instructed to complete a self-administered questionnaire via a Google Forms link accessible through a QR code. The final sample consisted of 267 participants.

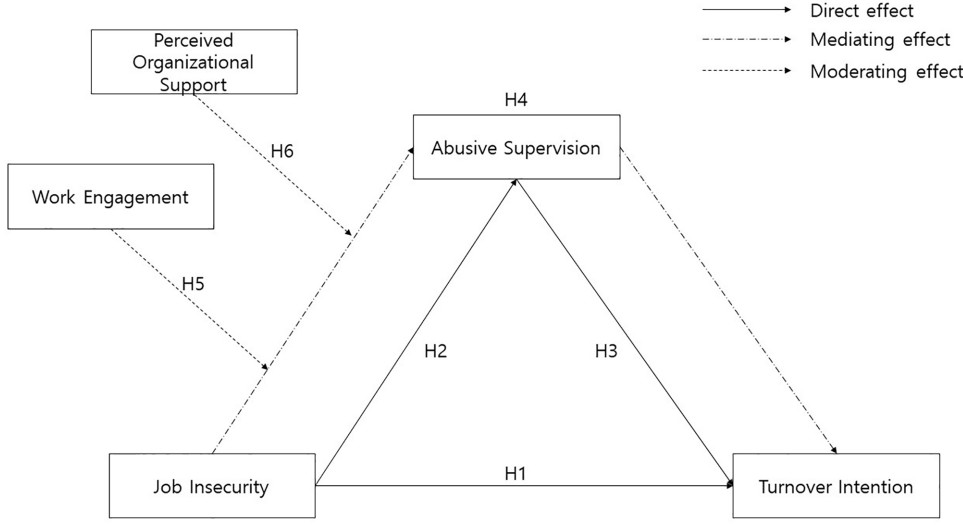

**Fig 1. Research model.**

Because participants were recruited through convenience sampling from selected gyms and sports centers, the sample may not fully represent all non-regular sports instructors in South Korea. Accordingly, the findings should be interpreted with caution in terms of population-level generalizability. This limitation may restrict the external validity of the study findings.

This study was approved by the Institutional Review Board of a public institution designated by the Ministry of Health and Welfare of Korea (Approval No. P01-202504-01-007). All procedures were conducted in accordance with relevant guidelines and regulations. Informed consent was obtained from all participants, and no identifiable personal information was collected.

## Measures

All constructs were measured using previously validated scales. Job insecurity was measured using Ashford et al.'s scale [58]. Abusive supervision was measured using Tepper's scale [13]. Work engagement was measured using the Utrecht Work Engagement Scale [15]. Perceived organizational support was measured using Eisenberger et al.'s scale [16]. Turnover intention was measured using Michaels and Spector's scale [59].

Respondents' demographic characteristics were measured using a total of five items (gender, age, education, tenure, and turnover experience). The key constructs of this study were defined as five main variables: job insecurity, abusive supervision, work engagement, perceived organizational support, and turnover intention. All survey items were structured on a 5-point Likert scale, where 1 indicated "Not at all" and 5 indicated "Very much." Each variable was adapted and revised based on validated tools from existing research to suit the study's objectives.

First, Job Insecurity Scale, which refers to the fear of losing one's job or the uncertainty about employment continuity causing psychological stress, was used in this study. The measurement tool was based on the theoretical definition and initial framework by Greenhalgh & Rosenblatt [32], further refined by Ashford et al. [58]. This scale conceptualizes job insecurity as a perceived threat and uncertainty regarding job sustainability, focusing on its relationship with psychological stress and organizational commitment. In South Korea, Huh et al. adapted this scale to suit hotel industry employees [60], and Hwang reconstructed the scale for school sports club instructors [61]. This study adjusted and supplemented items from these validated scales to fit the occupational environment of sports organization employees. The scale consists of five items, with higher scores indicating a stronger perception of threats to job sustainability.

Second, Abusive Supervision Scale refers to supervisory behaviors where leaders or superiors in an organization do not respect subordinates as individual and dignified beings, instead adopting an emotionally detached, authoritarian, and control-centered approach. This attitude, viewing members as instrumental means, can lead to various negative outcomes such as psychological withdrawal, decreased work engagement, and increased turnover intention. The measurement tool used in this study was based on the 20-item Leadership Behavior Assessment Scale by Schriesheim and Hinkin [62], further refined by Tepper [13]. Tepper's scale was developed to capture repetitive and continuous verbal and behavioral abuse by supervisors, allowing for structural measurement of negative leadership aspects. In South Korea, the scale was adapted to reflect the Korean organizational culture and emotional context [63–64]. These studies empirically analyzed the impact of authoritarian and undemocratic behaviors of supervisors on employees' emotions and job behaviors in the Korean workplace. This study modified and supplemented items from these prior studies to suit the occupational environment characteristics of sports organization employees. The scale consists of 15 items, with higher scores indicating a stronger negative perception of authoritarian and abusive supervisory control.

Third, Work Engagement Scale refers to a positive, fulfilling psychological state that individuals have toward their job, characterized by being deeply immersed in work, energetic, and feeling meaningfulness and passion. The measurement tool used in this study was based on the Utrecht Work Engagement Scale (UWES) developed by Schaufeli et al. [15], which was later refined by Schaufeli et al. [65]. In South Korea, UWES has been adapted to the Korean context [66–67],

verifying its validity and reliability before applying it to university administrative staff. This study modified and supplemented UWES's nine items (three items for each sub-factor) based on these prior studies to fit the job characteristics of sports organization employees. Higher scores indicate that the employee is energetic, finds meaning, and is engaged in their job.

Fourth, Perceived Organizations Scale (POS) refers to employees' general perception of the extent to which the organization recognizes their efforts and contributions and cares about their well-being [16]. Based on social exchange theory, the concept posits that the more the organization supports and cares for employees, the more positively employees will feel and commit to the organization. The measurement tool used in this study was based on the Perceived Organizational Support Scale developed by Eisenberger et al. [16]. This scale assesses how much the organization recognizes individual value and contributions and considers welfare, initially comprising 36 items but later used in various condensed forms (e.g., 8-item, 16-item versions). Eisenberger et al. verified that POS significantly relates to job satisfaction, organizational commitment, and turnover intention, strengthening POS's theoretical and empirical foundation [68]. In particular, they emphasized that organizational justice, supervisor support, and job autonomy significantly influence POS. Based on these prior studies, this study referred to the measurement framework by Eisenberger et al. [16,68], incorporating Kim and Lee's Korean application case to adjust items to the job environment [69]. The scale consists of eight items, with higher scores indicating a higher perception of organizational recognition and support.

Finally, the Turnover Intention Scale refers to the psychological inclination to leave the current organization, closely related to job satisfaction, organizational commitment, and perceptions of the job environment. The measurement tool used in this study was based on the theoretical definition and initial scale by Michaels and Spector [59]. Turnover intention is particularly significant in environments like sports organizations, where job stability is relatively low and work intensity is high. For example, in a study of public sports center employees, Kim confirmed that transformational leadership negatively impacts turnover intention, suggesting that leadership types can buffer turnover intention [70]. Additionally, Bae et al. conducted a meta-analysis on the relationship between organizational commitment and turnover intention in the sports workforce, proving that organizational commitment significantly reduces turnover intention [71]. This study adjusted and supplemented items from these validated scales to fit the occupational environment and context of sports organization employees. Turnover intention scale consists of five items, with higher scores indicating a stronger psychological tendency to leave the organization.

## Data analysis

Data were analyzed using SPSS 25 and AMOS 24. First, frequency analyses were conducted to summarize the demographic characteristics of the participants. Second, confirmatory factor analysis (CFA) was performed to assess the validity of the measurement model, including unidimensionality, convergent validity, and discriminant validity. Internal consistency reliability was evaluated using Cronbach's alpha. Third, correlation analyses were conducted to examine the associations among the study variables. Fourth, structural equation modeling (SEM) and Hayes' PROCESS macro Models 4 and 7 were used to test the hypothesized mediation and moderated mediation associations. To evaluate the robustness of the findings, additional analyses were conducted. Specifically, the main models were re-estimated after controlling for key demographic variables, including gender, age, tenure, education, and prior turnover experience. In addition, multicollinearity diagnostics were examined to ensure that the parameter estimates were not unduly influenced by overlap among predictors. Supplementary analyses were also performed to determine whether the substantive pattern of results remained consistent across alternative model specifications.

In addition, multicollinearity was assessed using variance inflation factor (VIF) values, and all values were below the recommended threshold of 5, indicating no multicollinearity concerns. To address potential common method bias, Harman's single-factor test was conducted. The results showed that a single factor did not account for the majority of the variance, suggesting that common method bias is unlikely to be a serious concern.

## 5. Results

### Descriptive statistics

Data were collected through 267 questionnaires. The sample size required to achieve a large effect size, as determined by a pre-test power analysis using G*Power version 3.1.9.7, was 89. Therefore, the sample size of 267 participants secured to test the research hypotheses was sufficient. Detailed descriptive statistics, including the sociodemographic information of the survey respondents, are presented in Table 1. The demographic characteristics of the final sample showed a gender distribution of 158 males (59.2%) and 109 females (40.8%). The age distribution was as follows: 20s - 142 (53.2%), 30s - 109 (40.8%), 40s and above – 16 (6.0%). Educational background comprised 106 high school graduates (39.7%), 86 graduate school graduates (32.2%), and 75 college graduates (28.1%). Length of service was classified as follows: Less than 1 year – 84 people (31.5%), 1–3 years – 75 people (28.1%), 3–5 years – 55 people (20.6%), 5–10 years – 30 people (11.2%), 10 years or more – 23 people (8.6%). Turnover intention experience, 105 (39.3%) had job-hopped, while 162 (60.7%) had not.

### Verification of validity and reliability

To assess the construct validity of the measurement items, CFA was conducted to evaluate both convergent validity and discriminant validity. As presented in Table 2, the fit indices of the measurement model exceeded all criteria for structural model fit proposed by Kline [72]: TLI > .90, CFI > .90, RMR < .08, SRMR < .10, and RMSEA < .10.

Convergent validity was verified through the average variance extracted (AVE) and composite reliability (CR) values. All variables used in this study exceeded the criteria set by Kim [73], which are AVE ≥ .5 and CR ≥ .7, indicating that all variables achieved convergent validity. Discriminant validity was confirmed by comparing the squared correlation coefficients with the AVE values; the AVE values were greater than the squared correlation coefficients, thus establishing discriminant validity (Table 2). Furthermore, the reliability analysis of all constructs used in this study resulted in values ranging from .870 to .964, confirming the reliability of all items [74].

**Table 1. Descriptive Statistics of study participants.**

| Variable | Category | N | % |
|---|---|---|---|
| Gender | Male | 158 | 59.2 |
|  | Female | 109 | 40.8 |
| Age | 20s | 142 | 53.2 |
|  | 30s | 109 | 40.8 |
|  | over 40s | 16 | 6.0 |
| Educational Background | High school graduate | 106 | 39.7 |
|  | Bachelor's | 75 | 28.1 |
|  | Master's | 86 | 32.2 |
| Length of Service | less than 1 year | 84 | 31.5 |
|  | 1–3 years | 75 | 28.1 |
|  | 3–5 years | 55 | 20.6 |
|  | 5–10 years | 30 | 11.2 |
|  | over 10 years | 23 | 8.6 |
| Turnover Intention Experience | Yes | 105 | 39.3 |
|  | No | 162 | 60.7 |
| Total |  | 267 | 100 |

Note. N = 267. Percentages may not sum to 100 due to rounding.

**Table 2. CFA Results for All Variables.**

| Construct and Item | | B | β | SE | t | AVE | CR | a |
|---|---|---|---|---|---|---|---|---|
| Job Insecurity (JI) | JI1 | 1 | .822 | | | .666 | .909 | .909 |
| | JI2 | .929 | .806 | .061 | 15.193*** | | | |
| | JI3 | .973 | .751 | .071 | 13.762*** | | | |
| | JI4 | 1.007 | .837 | .063 | 16.053*** | | | |
| | JI5 | 1.034 | .861 | .062 | 16.720*** | | | |
| Abusive Supervision (AS) | AS1 | 1 | .784 | | | .649 | .965 | .964 |
| | AS2 | .998 | .839 | .064 | 15.628*** | | | |
| | AS3 | 1.109 | .872 | .067 | 16.492*** | | | |
| | AS4 | 1.026 | .782 | .072 | 14.253*** | | | |
| | AS5 | 1.026 | .807 | .069 | 14.856*** | | | |
| | AS6 | 1.161 | .798 | .079 | 14.635*** | | | |
| | AS7 | 1.009 | .795 | .069 | 14.570*** | | | |
| | AS8 | 1.103 | .797 | .075 | 14.607*** | | | |
| | AS9 | 1.075 | .734 | .082 | 13.149*** | | | |
| | AS10 | 1.107 | .811 | .074 | 14.938*** | | | |
| | AS11 | 1.013 | .793 | .070 | 14.507*** | | | |
| | AS12 | 1.182 | .828 | .077 | 15.355*** | | | |
| | AS13 | 1.147 | .812 | .077 | 14.976*** | | | |
| | AS14 | .962 | .832 | .062 | 15.452*** | | | |
| | AS15 | 1.126 | .794 | .077 | 14.545*** | | | |
| Turnover Intention (TI) | TI1 | 1 | .889 | | | .730 | .931 | .931 |
| | TI2 | .989 | .920 | .043 | 22.986*** | | | |
| | TI3 | .905 | .877 | .044 | 20.673*** | | | |
| | TI4 | .773 | .752 | .050 | 15.458*** | | | |
| | TI5 | .859 | .823 | .047 | 18.174*** | | | |
| Work Engagement (WE) | Vitality1 | 1 | .887 | | | .695 | .871 | .870 |
| | Vitality2 | .893 | .692 | .067 | 13.247*** | | | |
| | Vitality3 | 1.072 | .906 | .051 | 20.835*** | | | |
| | Commitment1 | 1 | .886 | | | .764 | .907 | .923 |
| | Commitment2 | 1.083 | .901 | .050 | 21.516*** | | | |
| | Commitment3 | 1.001 | .835 | .054 | 18.419*** | | | |
| | Immersion1 | 1 | .789 | | | .629 | .836 | .871 |
| | Immersion2 | .998 | .843 | .069 | 14.537*** | | | |
| | Immersion3 | .840 | .745 | .067 | 12.601*** | | | |
| Perceived Organizational Support (POS) | POS1 | 1 | .846 | | | .761 | .962 | .962 |
| | POS2 | 1.167 | .919 | .056 | 20.899*** | | | |
| | POS3 | 1.181 | .874 | .062 | 19.005*** | | | |
| | POS4 | 1.099 | .866 | .059 | 18.661*** | | | |
| | POS5 | 1.121 | .903 | .056 | 20.172*** | | | |
| | POS6 | 1.065 | .868 | .057 | 18.740*** | | | |
| | POS7 | 1.132 | .895 | .057 | 19.855*** | | | |
| | POS8 | .971 | .805 | .059 | 16.463*** | | | |
| Model Fit | X² | df | p | Q | TLI | CFI | RMR | SRMR | RMSEA |
| | 1684.952 | 798 | .000 | 2.111 | .909 | .915 | .059 | .051 | .064 |

Note. TLI = Tucker-Lewis index, CFI = comparative fit index, RMR = root mean square residual, SRMR = standardized root mean square residual, RMSEA = root mean square error of approximation.

## Correlation analysis

To explore the fundamental relationships among the variables used in this study, a correlation analysis was conducted. As presented in Table 3, the results of the correlation analysis indicate that all correlation coefficients show statistically significant positive correlations, consistent with the hypothesized relationships among the variables in this study. Furthermore, all correlation coefficients were statistically significant and presented values less than 1, thereby securing discriminant validity among the constructs. Discriminant validity implies that measures of different constructs should display corresponding differences. This was tested using the methods proposed by Anderson and Gerbing [75] and Fornell and Larcker [76]. According to the correlation matrix ($\varphi$ matrix), a 95% confidence interval [correlation ± (2 × standard error)] should not include 1, and the AVE should be greater than the squared correlations between all constructs, which indicates discriminant validity (also evaluated by taking the square root of AVE values). The correlation coefficients among all variables ranged from.000 to.783, and the square root of the AVE values for latent factors ranged from.793 to.874, indicating that partial discriminant validity was secured. The analysis of all variables, as shown in Table 3, revealed significant differences among them. Therefore, discriminant validity was established through these verification methods. Furthermore, normality was verified for this study based on Kline's criteria [77]: absolute skewness ≤ 3 and absolute kurtosis ≤ 8.

## Direct effects of job insecurity, abusive supervision, and turnover intention

To test Hypotheses 1–3, a SEM analysis was conducted. The model's goodness-of-fit was evaluated, and the results of the model fit analysis are presented in Table 4.

The results of the model fit evaluation for the research model demonstrated acceptable fit indices. The absolute fit indices were $\chi^2 = 592.515$ ($df = 272$, $p < .001$), RMR = .060, RMSEA = .067, and SRMR = .046. The incremental fit indices were NFI = .902, TLI = .938, and CFI = .944. According to the criteria proposed by Vandenberg and Lance [78], a model can be considered acceptable when RMR is less than.05, RMSEA is below.08, and SRMR is below.10. Based on these standards, the model presented in this study is within acceptable parameters. Furthermore, Kim suggested that incremental indices such as TLI, CFI, and NFI should exceed.90 to indicate a good fit, which supports the model's suitability in the present analysis [73]. In order to examine the structural relationships among job insecurity, abusive supervision, and turnover intention by non-regular sports instructors, SEM was conducted. The findings are illustrated in Fig 2 and detailed in Table 5.

**Table 3. Correlation Analysis.**

| Category | | JI | AS | TI | VT | CM | IM | POS |
|---|---|---|---|---|---|---|---|---|
| JI | | **.816** | | | | | | |
| AS | | .486**( | **.806** | | | | | |
| TI | | .611** | .422** | **.854** | | | | |
| WE | VT | .000 | −.132* | −.201** | **.834** | | | |
| | CM | −.108 | −.169** | −.234** | .776** | **.874** | | |
| | IM | −.074 | −.155* | −.217** | .673** | .783** | **.793** | |
| POS | | −.487** | −.487** | −.495** | .366* | .456** | .419** | **.872** |
| Mean | | 2.41 | 1.58 | 2.60 | 3.50 | 3.87 | 3.75 | 3.69 |
| S.D. | | 1.03 | .76 | 1.17 | .86 | .85 | .82 | .92 |
| Skewness | | .49 | 1.86 | .11 | −.35 | −.69 | −.61 | −.62 |
| Kurtosis | | −.57 | 3.24 | −1.16 | −.21 | .65 | .50 | .22 |

Note. * $p < .05$, ** $p < .01$, Bold = $\sqrt{AVE}$, JI = Job Insecurity, AS = Abusive Supervision, TI = Turnover Intention, WE = Work Engagement, VT = Vitality, CM = Commitment, IM = Immersion, POS = Perceived Organization Support. Note. Diagonal values represent the square root of AVE.

**Table 4. Evaluation Results of the Research Model's Goodness-of-Fit.**

| χ² | df | p | cmin/df | RMR |
|---|---|---|---|---|
| 592.515 | .272 | .000 | 2.178 | .060 |
| NFI | TLI | CFI | RMSEA | SRMR |
| .902 | .933 | .944 | .067 | .046 |

Note. NFI = Normed Fit Index, TLI = Tucker-Lewis index, CFI = comparative fit index, SRMR = standardized root mean square residual, RMSEA = root mean square error of approximation.

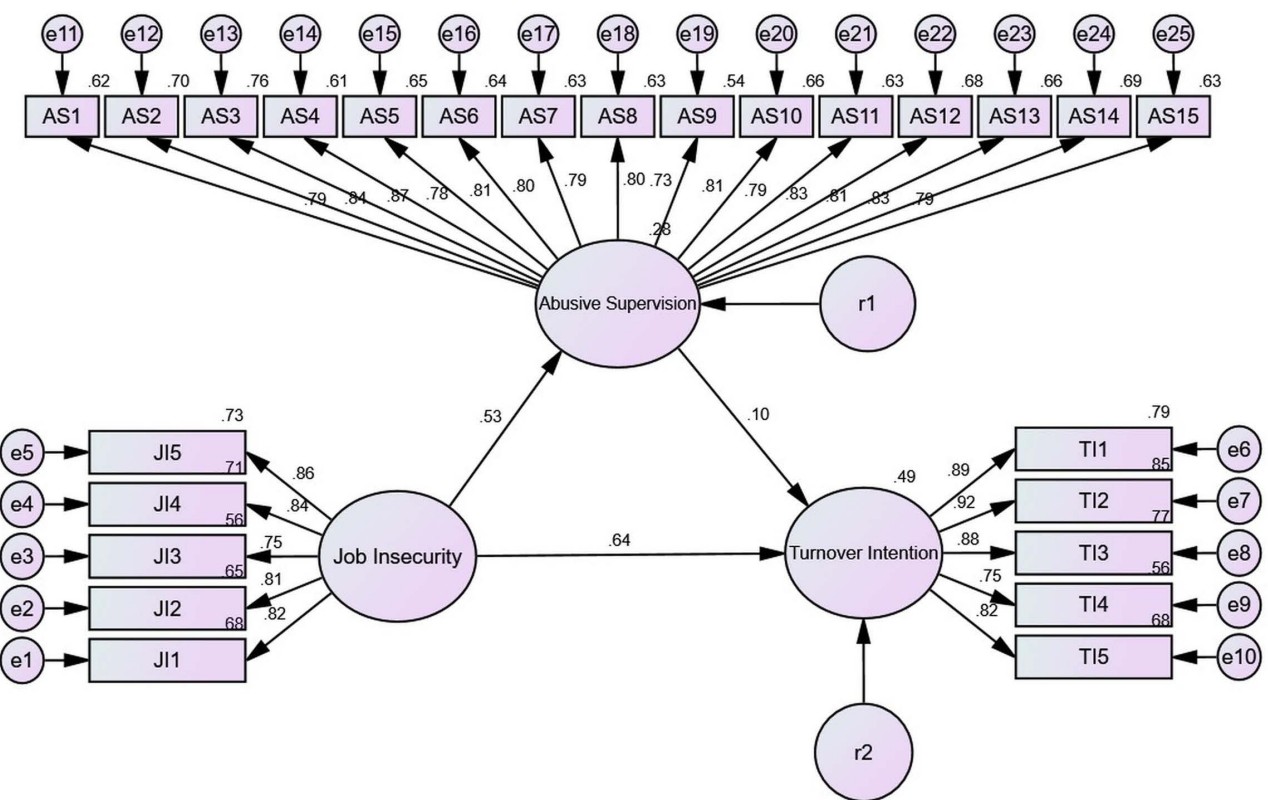

**Fig 2. Structural equation modeling for testing Hypotheses 1- 3.**

**Table 5. Results of the Path Analysis.**

| Hypotheses | Path | B | β | SE | CR | p | Result |
|---|---|---|---|---|---|---|---|
| H1 | JI→TI | .802 | .643 | .083 | 9.337 | .000 | Accepted |
| H2 | JI → AS | .368 | .525 | .046 | 7.984 | .000 | Accepted |
| H3 | AS → TI | .170 | .095 | .106 | 1.606 | .108 | Rejected |

Note. SE = standard error, CR = critical ratio; JI = job insecurity, AS = abusive supervision, TI = turnover intention.

The structural equation model analysis yielded the following results regarding the hypothesized relationships: H1 was supported, indicating that job insecurity had a significant and positive effect on turnover intention ($\beta = .643$, $p < .001$). H2 was also supported, demonstrating that job insecurity had a significant and positive impact on abusive supervision ($\beta = .525$, $p < .001$). H3 however, was not supported, as abusive supervision did not exhibit a statistically significant effect on turnover intention ($\beta = .095$, $p > .05$).

Interestingly, the path from abusive supervision to turnover intention was not statistically significant in the SEM analysis, whereas it was significant in the PROCESS analysis. This discrepancy may be attributed to differences in analytical approaches. SEM estimates all paths simultaneously within a full structural model, controlling for measurement error and shared variance among variables. In contrast, PROCESS macro relies on ordinary least squares regression and examines relationships in a more simplified model structure. Therefore, the significance observed in PROCESS may reflect direct effects that are not detected in the more stringent SEM framework.

### Mediating role of abusive supervision in the relationship between job insecurity and employees' turnover intention

To examine the mediating effect of abusive supervision on the relationship between job insecurity and turnover intention, an analysis was conducted using Process Macro Model 4 in SPSS. The results are presented in Table 6 below.

In the first stage, job insecurity was found to have a significant positive effect on abusive supervision ($\beta = .357$, $t = 9.054$, $p < .001$). In the second stage, job insecurity also significantly and positively affected turnover intention ($\beta = .601$, $t = 9.682$, $p < .001$). Additionally, abusive supervision significantly and positively influenced turnover intention ($\beta = .253$, $t = 2.997$, $p < .01$). In the third stage, the indirect effect of abusive supervision on the relationship between job insecurity and turnover intention was tested using bootstrapping with 5,000 resamples [79]. Following Hayes's criterion [80], an indirect effect is considered significant when the 95% confidence interval (CI) does not include zero. The mediation path showed a significant positive partial mediation effect ($\beta = .090$, 95% CI = [.041−.150]) because the confidence interval did not contain zero.

### The moderated mediation effect of work engagement on the mediating role of abusive supervision in the relationship between job insecurity and turnover intention

To examine the moderated mediation effect of work engagement on the mediating role of abusive supervision in the relationship between job insecurity and turnover intention, an analysis was conducted using Process Macro Model 7 in SPSS. The results are presented in Table 7.

**Table 6. The Mediating Effect of Abusive Supervision on the Relationship Between Job Insecurity and Turnover Intention.**

| Predictor | B | SE | t | p | LLCI | ULCI |
|---|---|---|---|---|---|---|
| Step 1. Effect of job insecurity on abusive supervision | | | | | | |
| JI | .357 | .039 | 9.054 | .000 | .280 | .435 |
| F = 82.969, R² = .236, p < .001 | | | | | | |
| Step 2. Effects of job insecurity and abusive supervision on turnover intention | | | | | | |
| JI | .601 | .062 | 9.682 | .000 | .479 | .723 |
| AS | .253 | .084 | 2.997 | .003 | .087 | .419 |
| F = 85.706, R² = .394, p < .001 | | | | | | |
| Step 3. Indirect effect | | | | | | |
| AS (indirect effect) | Effect | | Boot SE | Boot LLCI | Boot ULCI | |
| | .090 | | .028 | .041 | | .150 |

Note. PROCESS Model 4 was used with 5,000 bootstrap resamples; B = unstandardized coefficient, SE = standard error, LLCI = lower limit of the 95% confidence interval, ULCI = upper limit of the 95% confidence interval; JI = job insecurity, AS = abusive supervision.

**Table 7. The Moderated Mediation Effect of Work Engagement on the Mediating Role of Abusive Supervision in the Relationship Between Job Insecurity and Turnover Intention.**

| Predictor | B | SE | t | p | LLCI | ULCI |
|---|---|---|---|---|---|---|
| Step 1. Interaction term (JI × WE) predicting abusive supervision | | | | | | |
| Constant | 1.568 | .040 | 39.286 | .000 | 1.489 | 1.646 |
| JI | .373 | .039 | 9.443 | .000 | .295 | .450 |
| WE | −.158 | .053 | −2.993 | .000 | −.261 | −.054 |
| JI × WE | −.141 | .051 | −2.773 | .000 | −.241 | −.041 |
| F = 33.353, df1 = 3, df2 = 263, R² = .276, p < .001 | | | | | | |
| Step 2. Effects of job insecurity and abusive supervision on turnover intention | | | | | | |
| Constant | 2.201 | .144 | 15.253 | .000 | 1.917 | 2.486 |
| JI | .601 | .062 | 9.682 | .000 | .479 | .723 |
| AS | .253 | .084 | 2.997 | .003 | .087 | .419 |
| F = 85.706, df1 = 2, df2 = 264, R² = .394, p < .001 | | | | | | |
| Step 3. Conditional indirect effects at levels of work engagement | | | | | | |
| Index of moderated mediation | | Effect | Boot SE | Boot LLCI | Boot ULCI | |
| | | −.036 | .019 | −.075 | −.002 | |
| Work engagement level | Low (−1SD) | .481 | .061 | .361 | .601 | |
| | M | .373 | .039 | .295 | .450 | |
| | High (+1SD) | .264 | .050 | .166 | .362 | |

Note. PROCESS Model 7 was used with 5,000 bootstrap resamples. B = unstandardized coefficient; SE = standard error; LLCI = lower limit of the 95% confidence interval; ULCI = upper limit of the 95% confidence interval. JI = job insecurity; AS = abusive supervision; WE = work engagement.

The index of moderated mediation, representing the overall magnitude of the conditional indirect effect of job insecurity on turnover intention through abusive supervision as moderated by work engagement, was −0.036. The bootstrapped 95% confidence interval (95% CI = [−0.075, −0.002]) did not include zero, indicating a statistically significant moderated mediation effect.

To further interpret this effect, the conditional effect of job insecurity on abusive supervision was estimated at low (−1 SD), mean, and high (+1 SD) levels of work engagement. The results showed that the effect of job insecurity on abusive supervision was stronger at low levels of work engagement (B = 0.481) than at high levels of work engagement (B = 0.264). In addition, the effect size of the moderation effect was assessed using Cohen's f², indicating a small to moderate effect. This pattern indicates that the indirect effect of job insecurity on turnover intention through abusive supervision became weaker as work engagement increased, supporting the buffering role of work engagement in this mediated pathway.

**The moderated mediation effect of perceived organizational support on the mediating role of abusive supervision in the relationship between job insecurity and turnover intention**

To examine the moderated mediation effect of perceived organizational support on the mediating role of abusive supervision in the relationship between job insecurity and turnover intention, analysis was conducted using Process Macro Model 7 in SPSS. The results are presented in Table 8.

The index of moderated mediation, which represents the overall magnitude of the conditional indirect effect of job insecurity on turnover intention through abusive supervision as moderated by perceived organizational support, was −0.034. The bootstrapped 95% confidence interval (95% CI = [−0.089, −0.003]) did not include zero, indicating a statistically significant moderated mediation effect.

**Table 8. The Moderated Mediation Effect of Perceived Organizational Support on the Mediating Role of Abusive Supervision in the Relationship Between Job Insecurity and Turnover Intention.**

| Predictor | B | SE | t | p | LLCI | ULCI |
|---|---|---|---|---|---|---|
| Step 1. Interaction term (JI × POS) predicting abusive supervision | | | | | | |
| Constant | 1.514 | .041 | 37.041 | .000 | 1.433 | 1.594 |
| JI | .206 | .043 | 4.820 | .000 | .122 | .290 |
| POS | −.280 | .047 | −5.960 | .000 | −.372 | −.187 |
| JI × POS | −.133 | .035 | −3.834 | .000 | −.201 | −.065 |
| F = 33.353, df1 = 3, df2 = 263, R² = .354, p < .001 | | | | | | |
| Step 2. Effects of job insecurity and abusive supervision on turnover intention | | | | | | |
| Constant | 2.201 | .144 | 15.253 | .000 | 1.917 | 2.486 |
| JI | .601 | .062 | 9.682 | .000 | .479 | .723 |
| AS | .253 | .084 | 2.997 | .003 | .087 | .419 |
| F = 85.706, df1 = 2, df2 = 264, R² = .394, p < .001 | | | | | | |
| Step 3. Effects of job insecurity and abusive supervision on turnover intention | | | | | | |
| | | Effect | Boot SE. | Boot LLCI | | Boot ULCI |
| Index of Moderated Mediation | | −.034 | .022 | −.089 | | −.003 |
| Perceived organizational support level | Low (−1SD) | .328 | .048 | .234 | | .422 |
| | M | .206 | .234 | .122 | | .290 |
| | High (+1SD) | .083 | .058 | −.032 | | .198 |

Note. PROCESS Model 7 was used with 5,000 bootstrap resamples. B = unstandardized coefficient; SE = standard error; LLCI = lower limit of the 95% confidence interval; ULCI = upper limit of the 95% confidence interval. JI = job insecurity; AS = abusive supervision; TI = turnover intention; POS = perceived organizational support.

To further probe this effect, the conditional effect of job insecurity on abusive supervision was estimated at low (−1 SD), mean, and high (+1 SD) levels of perceived organizational support. The results showed that the effect of job insecurity on abusive supervision was stronger at low levels of perceived organizational support (B = 0.328) than at high levels of perceived organizational support (B = 0.083). In addition, the effect size of the moderation effect was assessed using Cohen's f², indicating a small to moderate effect. Accordingly, the indirect effect of job insecurity on turnover intention through abusive supervision was attenuated at higher levels of perceived organizational support, indicating that perceived organizational support functions as a buffering organizational resource.

## 6. Discussion

This study empirically analyzed the effect of job insecurity on turnover intention among non-regular sports instructors and verified the mediating effect of abusive supervision and the moderated mediating effects of work engagement and perceived organizational support. Based on the validated results, the following discussions are presented in conjunction with prior studies.

First, to analyze the effects of job insecurity on abusive supervision and turnover intention, a SEM analysis was conducted. The results revealed that job insecurity had a significant positive effect on both turnover intention and abusive supervision. However, abusive supervision did not have a significant direct effect on turnover intention. The finding that job insecurity positively affects turnover intention suggests that increased job insecurity is associated with greater intention to leave. This aligns with previous studies indicating that organizational members experiencing job insecurity tend to exhibit higher turnover intentions [45,46,81–83]. In particular, non-regular employees have been reported to exhibit higher turnover intentions than regular employees due to unstable employment conditions [84]. Since non-regular sports instructors,

the subjects of this study, face structural job insecurity [20], it is reasonable to interpret these findings as reflecting a heightened tendency toward turnover. From this perspective, improving the unstable work environments that non-regular workers in the sports industry face is crucial to reducing their turnover intentions.

Regarding the result of job insecurity having a significant positive effect on abusive supervision, this implies that a high level of job insecurity may lead to deteriorated relationships with supervisors. This is consistent with research suggesting that supervisors' job insecurity can increase abusive supervision behaviors [85]. An unstable employment environment may increase supervisors' awareness of their own job insecurity, which, in turn, negatively affects their behavior toward subordinates [86]. These results suggest that job insecurity not only affects individual turnover intentions but also damages the relationships between supervisors and subordinates, including those between sports instructors.

In this study, abusive supervision was not found to have a direct effect on turnover intention. This is consistent with Choi's findings [87], indicating no direct relationship between abusive supervision and turnover intention. Nevertheless, abusive supervision is widely regarded as a factor associated with negative outcomes for individuals and organizations [88]. Prior studies have reported a significant association between abusive supervision and turnover intention [89–92]. Such inconsistent findings in the literature suggest that the relationship between abusive supervision and turnover intention may vary depending on organizational context or situational factors.

Second, using Hayes Process Macro Model 4, the mediating effect of abusive supervision in the relationship between job insecurity and turnover intention was examined. The analysis revealed a significant positive mediating effect. In a similar context, Qian et al. provided empirical evidence that leadership style mediates the relationship between job insecurity and turnover intention [42]. Supervisory styles directly impact communication and emotional bonds within an organization, and abusive supervision undermines these bonds, thereby serving as a negative factor that can promote turnover [13]. In sports organizations, where hierarchical power structures, short-term contracts, and emotional labor intensity are common, abusive supervision is frequently observed [40]. In such environments, higher levels of job insecurity may intensify abusive supervisory behaviors, which can indirectly increase turnover intention.

Third, the moderated mediating effect of work engagement on the relationship between job insecurity and turnover intention via abusive supervision was tested using Hayes Process Macro Model 7. The analysis revealed a conditional indirect effect, indicating a negative moderated mediating effect of work engagement. Specifically, the mediating effect of abusive supervision was stronger in groups with low work engagement. By contrast, the indirect effect was weaker among individuals with high work engagement. This finding indicates that work engagement functions as a psychological buffer that attenuates the extent to which job insecurity increases turnover intention through abusive supervision. In other words, non-regular sports instructors with higher work engagement appear less vulnerable to the negative indirect pathway linking job insecurity, abusive supervision, and turnover intention. Park and Kim found that work engagement serves as a psychological buffering variable that moderates the indirect relationship between job insecurity and turnover intention, which is consistent with the present findings [93].

Fourth, the moderated mediating effect of perceived organizational support in the relationship between job insecurity and turnover intention via abusive supervision was analyzed using Hayes Process Macro Model 7. The results indicated a conditional indirect effect, with perceived organizational support demonstrating a negative moderated mediating effect. This suggests that the mediating effect of abusive supervision is stronger among individuals who perceive lower organizational support, whereas the indirect effect is weaker among those who perceive higher organizational support. This finding suggests that perceived organizational support functions as an organizational resource that mitigates the adverse indirect effect of job insecurity on turnover intention through abusive supervision. Employees who perceive stronger organizational support appear less susceptible to this negative pathway. These findings align with previous research [16], which emphasized that perceived organizational support strengthens emotional connections with the organization and reduces turnover intention. Therefore, when perceived organizational support is high, the pathway from job insecurity through abusive supervision to turnover intention may be weakened.

This study should be interpreted in light of several limitations. First, the use of convenience sampling and voluntary participation may have introduced sampling bias. Second, because the sample was limited to non-regular sports instructors recruited from selected sites in South Korea, the external validity of the findings is necessarily constrained. Therefore, caution is warranted when generalizing these results to other occupational groups, regions, or national contexts.

## 7. Conclusions

This study empirically examined the impact of perceived job insecurity on turnover intention among non-regular sports instructors, focusing on the mediating role of abusive supervision and the moderated mediation effects of work engagement and perceived organizational support. Specifically, the research aimed to identify the pathway through which job insecurity leads to turnover intention via abusive supervision and to explore how work engagement and organizational support perceptions can mitigate turnover intentions. Ultimately, the study sought to provide practical implications for alleviating job insecurity and reducing turnover intention, thereby addressing emotional labor challenges faced by non-regular sports instructors.

First, consistent with Hypotheses 1 and 2, perceived job insecurity among non-regular sports instructors was found to have a direct positive effect on turnover intention and abusive supervision. This finding indicates that heightened job insecurity directly stimulates turnover intention and exacerbates negative supervisor-subordinate dynamics, potentially contributing to an organizational climate marked by authoritative or emotionally detached leadership.

Second, Hypothesis 3 revealed that abusive supervision did not exert a statistically significant direct influence on turnover intention. This result contrasts with some prior studies and may reflect the unique structural characteristics of sports organizations, such as the high proportion of contract workers and vertical organizational culture. It suggests that non-regular sports instructors may respond more sensitively to structural factors such as job continuity rather than supervisory behaviors alone.

Third, in line with Hypothesis 4, abusive supervision significantly mediated the relationship between job insecurity and turnover intention. This underscores the importance of leadership development and management improvement at the organizational level to prevent negative interpersonal interactions that amplify turnover risks among non-regular sports instructors.

Fourth, Hypothesis 5 confirmed that work engagement functions as a moderator in the indirect pathway from job insecurity to turnover intention. Non-regular sports instructors with high levels of work engagement appear more resilient to the adverse effects of job insecurity and exhibit greater psychological resistance to abusive supervision. This supports the view from positive psychology that internal motivation and immersion can buffer organizational stress.

Fifth, Hypothesis 6 demonstrated that perceived organizational support also plays a critical moderating role in the indirect effect of job insecurity on turnover intention. Specifically, higher levels of perceived organizational support attenuate the pathway from abusive supervision to turnover intention, emphasizing the vital role of emotional and tangible organizational support in turnover prevention.

Collectively, these findings suggest that reducing turnover intention among non-regular sports instructors requires multifaceted organizational efforts. Beyond improving employment conditions and contract stability, fostering psychological safety, enhancing leadership styles, and strengthening organizational support systems are essential. Given that job insecurity can trigger detrimental interpersonal dynamics that escalate turnover intention, leadership training and organizational culture reforms must be prioritized. Furthermore, promoting work engagement and ensuring that employees perceive strong organizational support represent critical intervention points that can sustainably improve job satisfaction and retention.

This study contributes to the theoretical framework of turnover intention among non-regular sports workers and offers actionable insights for organizational behavior and sports management scholarship. Future research should consider incorporating variables such as conflict management, emotional labor buffering factors, and LMX relationships to develop more nuanced explanatory models.

 

## Recommendations for future studies

Based on the findings of the present study, the following directions for future research are proposed.

First, this study empirically identified the mediating effect of abusive supervision and the moderated mediation effects of work engagement and perceived organizational support on the relationship between job insecurity and turnover intention among non-regular sports instructors. However, considering the multifaceted nature of job insecurity, future research should explore additional internal and external variables related to job stability. This would facilitate the development of comprehensive and concrete strategies to mitigate the adverse impact of job insecurity on turnover intention. Particularly, employing qualitative methodologies alongside quantitative approaches could provide deeper insights into the lived experiences, working environments, and psychological states of non-regular sports instructors, thereby enriching the understanding of their occupational challenges.

Second, this study treated various job categories within the sports domain collectively under the single category of "sports instructors." However, the sports instructor population comprises diverse subgroups that differ in job characteristics, employment conditions, and organizational contexts. Future research should therefore segment these subgroups more precisely and conduct comparative analyses to better elucidate the relationship between job insecurity and turnover intention. Such segmentation would not only clarify the unique demands and challenges of each subgroup but also enable the formulation of tailored interventions aimed at enhancing job stability and reducing turnover intention within each distinct group.

Third, the findings highlight the necessity of proactive organizational interventions to improve job stability for non-regular sports instructors. Future studies should examine the real-world effects and changes resulting from the implementation of organizational policies, systems, and support programs. Specifically, longitudinal assessments of job satisfaction, job security, and turnover intention following organizational interventions are crucial. These evaluations would empirically verify the extent to which organizational roles and responsibilities contribute to alleviating job insecurity and curbing workforce attrition. Ultimately, such research will provide evidence-based guidance for policy and practice aimed at fostering sustainable employment conditions and stable organizational operation within the sports industry.

## Supporting information

**S1 File. Dataset used in this study.** The dataset is available at: https://osf.io/92wsf/.
(XLSX)

## Author contributions

**Conceptualization:** Kwon-Hyuk Jeong, Heesu Mun.

**Data curation:** Geon-Ha Jeong.

**Formal analysis:** Geon-Ha Jeong.

**Funding acquisition:** Kwon-Hyuk Jeong.

**Investigation:** Geon-Ha Jeong, Heesu Mun.

**Methodology:** Kwon-Hyuk Jeong, Geon-Ha Jeong.

**Project administration:** Kwon-Hyuk Jeong, Heesu Mun.

**Resources:** Kwon-Hyuk Jeong.

**Software:** Geon-Ha Jeong.

**Supervision:** Kwon-Hyuk Jeong.

**Validation:** Heesu Mun.

**Visualization:** Heesu Mun.

**Writing – original draft:** Heesu Mun.

**Writing – review & editing:** Kwon-Hyuk Jeong.

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
