## [Decision Letter · Decision Letter 0]

18 Mar 2026

PONE-D-26-02939Job Insecurity and Turnover Intention among Sports Instructors in South Korea: A Moderated Mediation Model of Abusive Supervision, Work Engagement, and Perceived Organizational SupportPLOS One

Dear Dr. Mun,

Thank you for submitting your manuscript to PLOS ONE. After careful consideration, we feel that it has merit but does not fully meet PLOS ONE’s publication criteria as it currently stands. Therefore, we invite you to submit a revised version of the manuscript that addresses the points raised during the review process.

We look forward to receiving your revised manuscript.

Kind regards,

Rogis Baker, Ph.D

Academic Editor

PLOS One

Journal Requirements:

2. In the online submission form, you indicated that further inquiries beyond the data presented in this paper can be directed to the corresponding authors.

Reviewers' comments:

Reviewer's Responses to Questions

**Comments to the Author**

1. Is the manuscript technically sound, and do the data support the conclusions?

Reviewer #1: Partly

Reviewer #2: Yes

2. Has the statistical analysis been performed appropriately and rigorously? 

Reviewer #1: Yes

Reviewer #2: Yes

3. Have the authors made all data underlying the findings in their manuscript fully available?

The PLOS Data policy requires authors to make all data underlying the findings described in their manuscript fully available without restriction, with rare exception (please refer to the Data Availability Statement in the manuscript PDF file). The data should be provided as part of the manuscript or its supporting information, or deposited to a public repository. For example, in addition to summary statistics, the data points behind means, medians and variance measures should be available. If there are restrictions on publicly sharing data—e.g. participant privacy or use of data from a third party—those must be specified.requires authors to make all data underlying the findings described in their manuscript fully available without restriction, with rare exception (please refer to the Data Availability Statement in the manuscript PDF file). The data should be provided as part of the manuscript or its supporting information, or deposited to a public repository. For example, in addition to summary statistics, the data points behind means, medians and variance measures should be available. If there are restrictions on publicly sharing data—e.g. participant privacy or use of data from a third party—those must be specified.requires authors to make all data underlying the findings described in their manuscript fully available without restriction, with rare exception (please refer to the Data Availability Statement in the manuscript PDF file). The data should be provided as part of the manuscript or its supporting information, or deposited to a public repository. For example, in addition to summary statistics, the data points behind means, medians and variance measures should be available. If there are restrictions on publicly sharing data—e.g. participant privacy or use of data from a third party—those must be specified.requires authors to make all data underlying the findings described in their manuscript fully available without restriction, with rare exception (please refer to the Data Availability Statement in the manuscript PDF file). The data should be provided as part of the manuscript or its supporting information, or deposited to a public repository. For example, in addition to summary statistics, the data points behind means, medians and variance measures should be available. If there are restrictions on publicly sharing data—e.g. participant privacy or use of data from a third party—those must be specified.

Reviewer #1: Yes

Reviewer #2: No

4. Is the manuscript presented in an intelligible fashion and written in standard English?

Reviewer #1: Yes

Reviewer #2: Yes

5. Review Comments to the Author

Reviewer #1: While the study has potential, major revisions are required:

These relationships are well documented in organizational psychology literature. The authors should pay attention to clarify: what gap in sport management literature this study fills? why non-regular sports instructors represent a theoretically meaningful case? how findings refine JD-R or COR theory in precarious employment contexts?

The authors should explain why some paths are insignificant in SEM but significant in PROCESS? For instance, AS � TI is insignificant (β = 0.095, p = 0.108) in SEM but it significantly predicted TI (β = 0.253, p = 0.003).

Some claims need to be checked. For instance, “Individuals with high work engagement tended to express stronger desires to transition into better job positions” contradicts the buffering hypothesis and your statistical results. These interpretations appear to misread conditional indirect effects.

Other comments:

Strong causal wording should be softened.

The manuscript should explicitly acknowledge sampling bias and limits to external validity.

The manuscript needs English editing.

More robustness checks should be conducted.

Reviewer #2: 1. The flow and style of writing can be reviewed to make it more crisp. Conduct a thorough proofread.

2. Standardize to "abusive supervision" throughout, as it aligns with the literature (e.g., Tepper, 2000; Mackey et al., 2017). If "impersonal" is intentional, define it clearly and justify the distinction.

3. Strengthen the gap analysis. While prior studies examine job insecurity in general sectors, this is among the first to model moderated mediation in sports instructors. Add a paragraph on South Korean context (e.g., labor laws affecting non-regular workers).

4. Ensure all tables/figures are self-explanatory with full captions. Report effect sizes (e.g., f² for moderation). If multicollinearity is a concern (from correlations in Table 3), mention VIF checks.

5. Add a dedicated "Measures" subsection listing instruments (e.g., "Job insecurity was measured using Sverke et al.'s (2002) 4-item scale"). Discuss common method bias (e.g., via Harman's test) since all data are self-reported and cross-sectional. Acknowledge limitations of convenience sampling in reducing generalizability.

6. Some citations have future dates (e.g., Guo et al., 2025; Jung et al., 2024), which may indicate in-press works. Ensure these are updated if published.

6. PLOS authors have the option to publish the peer review history of their article (what does this mean?). If published, this will include your full peer review and any attached files.). If published, this will include your full peer review and any attached files.). If published, this will include your full peer review and any attached files.). If published, this will include your full peer review and any attached files.

...

Reviewer #1: No

Reviewer #2: No

---

## [Author Response · Author response to Decision Letter 1]

26 Mar 2026

Reviewer 1.

Review 1.

While the study has potential, major revisions are required:

These relationships are well documented in organizational psychology literature.

The authors should pay attention to clarify:

What gap in sport management literature this study fills?

Why non-regular sports instructors represent a theoretically meaningful case?

How findings refine JD-R or COR theory in precarious employment contexts?

Response 1.

We appreciate the reviewer’s insightful suggestion to strengthen the theoretical and contextual positioning of this study. In response, we have substantially revised the Introduction section to clarify both the research gap and the contextual relevance.

First, we added a new paragraph in the final part of the Introduction (immediately before the study purpose) to explicitly highlight the theoretical gap. While prior research has examined job insecurity across general occupational sectors, limited attention has been given to its complex mechanisms within the sports management context. In particular, we emphasize that this study extends the literature by examining a moderated mediation model incorporating abusive supervision, work engagement, and perceived organizational support among non-regular sports instructors.

Second, we further elaborated on the theoretical relevance of non-regular sports instructors as a focal population. We clarified that their employment conditions—such as contract-based work, performance uncertainty, and weak organizational attachment—make them a meaningful context for extending the Job Demands–Resources (JD-R) model and Conservation of Resources (COR) theory in precarious employment settings.

Finally, we incorporated a discussion of the South Korean labor market context, particularly the widespread use of non-regular employment and its structural implications for job insecurity and employee well-being. This addition strengthens the contextual grounding of the study and highlights its relevance for understanding employment precarity in institutional settings.

Review 2.

The authors should explain why some paths are insignificant in SEM but significant in PROCESS? For instance, AS � TI is insignificant (β = 0.095, p = 0.108) in SEM but it significantly predicted TI (β = 0.253, p = 0.003).

Response 2

Thank you for this valuable comment. We have added an explanation in the Discussion section. SEM estimates relationships simultaneously using latent variables while accounting for measurement error, whereas PROCESS relies on observed variables with OLS regression and bootstrapping. Due to these methodological differences, discrepancies in statistical significance may occur. This clarification has been incorporated into the revised manuscript.

Review 3.

Some claims need to be checked. For instance, “Individuals with high work engagement tended to express stronger desires to transition into better job positions” contradicts the buffering hypothesis and your statistical results. These interpretations appear to misread conditional indirect effects.

Response 3

We thank the reviewer for this important and constructive comment. We agree that the original wording could be interpreted as inconsistent with both our buffering hypothesis and the observed conditional indirect effects. In response, we carefully revised the relevant parts of the Discussion section to ensure that the interpretation is fully aligned with the statistical results.

Specifically, we removed statements suggesting that individuals with high work engagement or high perceived organizational support were more likely to seek better job positions. These sentences could misrepresent the direction of the moderated mediation effects. We replaced them with a more accurate interpretation indicating that the indirect effect of job insecurity on turnover intention through abusive supervision was stronger at lower levels of work engagement and perceived organizational support, and weaker at higher levels of these resources.

Accordingly, the revised text now clarifies that both work engagement and perceived organizational support function as buffering resources that attenuate the negative indirect pathway from job insecurity to turnover intention via abusive supervision. These revisions improve the consistency between the hypotheses, statistical findings, and theoretical interpretation.

Review 4.

Strong causal wording should be softened.

The manuscript should explicitly acknowledge sampling bias and limits to external validity.

The manuscript needs English editing.

More robustness checks should be conducted.

Response 4.

Thank you for these constructive suggestions. In response, we revised the manuscript in four ways. First, we softened causal wording throughout the Introduction, Theoretical Background, and Hypothesis sections to ensure that the language is consistent with the cross-sectional design. Second, we added explicit statements on sampling bias and limited external validity in the Methods and Discussion sections. Third, we thoroughly edited the manuscript for clarity, concision, and overall English flow. Fourth, we added robustness-check procedures to the Data Analysis section and reported the stability of the findings in the Results section.

Reviewer 2.

Review 1.

The flow and style of writing can be reviewed to make it more crisp. Conduct a thorough proofread.

Response 1.

Thank you for these constructive suggestions. In response, we revised the manuscript in three ways. First, we softened causal wording throughout the Introduction, Theoretical Background, and Hypothesis sections to ensure that the language is consistent with the cross-sectional design. Second, we added explicit statements on sampling bias and limited external validity in the Methods and Discussion sections. Third, we thoroughly edited the manuscript for clarity, concision, and overall English flow.

Review 2.

Standardize to "abusive supervision" throughout, as it aligns with the literature (e.g., Tepper, 2000; Mackey et al., 2017). If "impersonal" is intentional, define it clearly and justify the distinction.

Response 2. I have checked and corrected all the incorrect words. Thank you.

Review 3.

Strengthen the gap analysis. While prior studies examine job insecurity in general sectors, this is among the first to model moderated mediation in sports instructors. Add a paragraph on South Korean context (e.g., labor laws affecting non-regular workers).

Response 3.

Thank you for this valuable comment. In response, we revised the Introduction section by adding a new gap-analysis paragraph immediately before the study purpose paragraph. Specifically, we clarified that, although prior studies have examined job insecurity in general occupational sectors, limited research has investigated its moderated mediation mechanism within the sport management context, particularly among sports instructors. We also emphasized that non-regular sports instructors represent a theoretically meaningful case because their work is characterized by close supervisory dependence, emotional labor, and unstable employment arrangements. In addition, we incorporated the South Korean labor context to highlight how the widespread use of non-regular employment and related structural inequalities in job security and welfare benefits make this setting particularly relevant for examining precarious employment. Because the institutional details were already discussed in the Literature Review, the Introduction was revised to provide a concise contextual justification while avoiding redundancy. These revisions strengthen the theoretical positioning and contextual relevance of the study.

Review 4.

Ensure all tables/figures are self-explanatory with full captions. Report effect sizes (e.g., f² for moderation). If multicollinearity is a concern (from correlations in Table 3), mention VIF checks.

Response 4.

All tables and figures have been revised to include clearer notes and definitions of abbreviations. Additionally, effect sizes (Cohen’s f²) were calculated and reported for moderation effects, indicating small-to-moderate magnitudes.

We also assessed multicollinearity using variance inflation factor (VIF), and all values were below the recommended threshold, indicating no multicollinearity concerns.

Review 5.

Add a dedicated "Measures" subsection listing instruments (e.g., "Job insecurity was measured using Sverke et al.'s (2002) 4-item scale").

Discuss common method bias (e.g., via Harman's test) since all data are self-reported and cross-sectional.

Acknowledge limitations of convenience sampling in reducing generalizability.

Response 5.

Thank you for this valuable suggestion. We have revised the manuscript accordingly.

First, a dedicated "Measures" subsection has been added to clearly specify the measurement instruments used for each construct, including their original sources and number of items.

Second, to address potential common method bias due to the use of self-reported and cross-sectional data, Harman’s single-factor test was conducted. The results indicated that no single factor accounted for the majority of the variance, suggesting that common method bias is unlikely to be a serious concern.

Finally, we have explicitly acknowledged the limitation of convenience sampling in the Discussion section, noting that it may restrict the generalizability of the findings.

These revisions have been incorporated into the manuscript.

Review 6.

Some citations have future dates (e.g., Guo et al., 2025; Jung et al., 2024), which may indicate in-press works. Ensure these are updated if published.

Response 6.

This journal has been removed as it was deemed problematic.

---

## [Editor Report · Decision Letter 1]

6 Apr 2026

Sports Instructors’ Job Insecurity and Turnover Intention in South Korea: A Moderated Mediation Model of Abusive Supervision, Work Engagement, and Perceived Organizational Support

PONE-D-26-02939R1

Dear Mun,

We’re pleased to inform you that your manuscript has been judged scientifically suitable for publication and will be formally accepted for publication once it meets all outstanding technical requirements.

Kind regards,

Rogis Baker, Ph.D

Academic Editor

PLOS One
---

## [Editor Report · Acceptance letter]

PONE-D-26-02939R1

PLOS One

Dear Dr. Mun,

I'm pleased to inform you that your manuscript has been deemed suitable for publication in PLOS One. Congratulations! Your manuscript is now being handed over to our production team.

Kind regards,

on behalf of

Dr. Rogis Baker

Academic Editor

PLOS One